# Drug Resistance and Molecular Characteristics of *Mycobacterium tuberculosis*: A Single Center Experience

**DOI:** 10.3390/jpm12122088

**Published:** 2022-12-19

**Authors:** Shanshan Li, Wen Chen, Mengru Feng, Yuejiao Liu, Fenghua Wang

**Affiliations:** Department of Pathology, The Eighth Medical Center, Chinese PLA General Hospital, Beijing 100091, China

**Keywords:** *Mycobacterium tuberculosis*, drug-resistant, gene mutation, molecular characteristic, PCR

## Abstract

In recent years, the incidence of tuberculosis (TB) and mortality caused by the disease have been decreasing. However, the number of drug-resistant tuberculosis patients is increasing rapidly year by year. Here, a total of 380 *Mycobacterium tuberculosis* (MTB)-positive formalin-fixed and paraffin-embedded tissue (FFPE) specimens diagnosed in the Department of Pathology of the Eighth Medical Center, Chinese PLA General Hospital were collected. Among 380 cases of MTB, 85 (22.37%) were susceptible to four anti-TB drugs and the remaining 295 (77.63%) were resistant to one or more drugs. The rate of MDR-TB was higher in previously treated cases (52.53%) than in new cases [(36.65%), *p*  <  0.05]. Of previously treated cases, the rate of drug resistance was higher in females than in males (*p*  <  0.05). Among specimens obtained from males, the rate of drug resistance was higher in new cases than in previously treated cases (*p*  <  0.05). Of mutation in drug resistance-related genes, the majority (53/380, 13.95%) of *rpoB* gene carried the D516V mutation, and 13.42% (51/380) featured mutations in both the *katG* and *inhA* genes. Among the total specimens, 18.68% (71/380) carried the 88 M mutation in the *rpsL* gene, and the *embB* gene focused on the 306 M2 mutation with a mutation rate of 19.74%. Among the resistant INH, the mutation rate of −15 M was higher in resistance to more than one drug than in monodrug-resistant (*p*  <  0.05). In conclusion, the drug resistance of MTB is still very severe and the timely detection of drug resistance is conducive to the precise treatment of TB.

## 1. Introduction

Tuberculosis (TB) disease is a chronic infectious disease caused by *Mycobacterium tuberculosis* (MTB). Globally, there were an estimated 9.9 million new cases of MTB in 2020 [1], and about half a million of these cases were multidrug-resistant TB (MDR-TB) [2]. The emergency of drug-resistant TB (DR-TB) has threatened global public health efforts to control TB [3,4]. China has the third-highest burden of MTB in the world [5], in particular a significant increase in DR-TB, which severely impacts treatment outcomes [6]. The timely diagnosis and treatment of TB, particularly DR-TB, contributes to controlling the transmission of the disease [2]. Acid-fast staining and molecular pathology tests are the primary diagnostic methods for MTB infection. The health industry standard “WS 288-2017 Diagnosis for pulmonary tuberculosis” published by the PRC National Health and Family Planning Commission clearly identifies the important role that molecular pathology plays in the diagnosis and treatment of drug-resistant MTB (DR-MTB) [7].

Rifampicin (RIF), isoniazid (INH), streptomycin (STR), and ethambutol (EMB) are four first-line anti-MTB drugs [8] - that is a double-edged sword. While eliminating pathogenic MTB, they also select drug-resistant bacteria against which the drugs are useless [9]. Resistance to anti-MTB drugs is obtained mainly through mutations in distinctive drug resistance-related genes [10,11], which mainly include *rpoB*, *inhA*, *katG*, *rpsL*, and *embB* [11]. The molecular mechanisms by which MTB develops resistance to each drug are different.

RIF inhibits MTB replication by interacting with the RNA polymerase β subunit encoded by the *rpoB* gene [12]. The majority of rifampicin-resistant tuberculosis (RR-TB) carries a mutation in an 81-bp RFP resistance-determining region (RRDR) of the *rpoB* gene [13]. *katG* gene inhibits mycolic acids synthesis in the MTB cell wall by encoding a hydrogen peroxide-peroxidase that oxidizes INH to isonicotinic acid and by participating in NAD synthesis, disrupting its antioxidant and anti-invasive functions to kill bacteria [14]. The protein encoded by the *inhA* gene is a NADH-enoyl acetyl carrier protein reductase (InhA), which is related to mycolic acids biosynthesis and catalyzes the formation of mycolic acids from short-chain fatty acid precursors [15]. STR inhibits protein synthesis by irreversibly binding to ribosomal protein S12 and 16S rRNA, interfering with translation proofreading [16]. Mutations in *rpsL*, the gene encoding ribosomal protein S12, can lead to resistance of MTB to STR [17]. EMB inhibits mycobacterium arabinosyl transferases encoded by the *embCAB* operon, affecting the formation of mycobacterium acid-arabinogalactan-proteoglycan in the cell wall of MTB, which is essential for maintaining cell structure [18]. Codon 306 of the *embB* gene is the most common point mutation detected in clinical isolates of EMB-resistant MTB [19].

Understanding the drug resistance and molecular characteristics of MTB to the four first-line drugs is helpful for the treatment and epidemiological study of MTB. Polymerase chain reaction (PCR)-reverse membrane hybridization technology used to assay 380 MTB-positive formalin-fixed and paraffin-embedded tissue (FFPE) specimens. Summary and analysis of the drug resistance and molecular characteristics, provide a theoretical reference for the treatment of MTB.

## 2. Materials and Methods

### 2.1. Specimen Collection

Laboratory examinations were performed in the molecular pathology laboratory of the Department of Pathology of the Eighth Medical Center, Chinese PLA General Hospital, an ISO 15189 accredited laboratory specialized in MTB detection. There were 380 cases of MTB-positive FFPE specimens collected from January 2016 to December 2020. All samples had been confirmed to contain MTB DNA through *Mycobacterium* species identification experiment. Acid-fast staining and H&E staining was also performed to observe the histopathological changes. The basic information of the specimens is shown in Table 1.

### 2.2. Drug Resistance Pattern

Monodrug-resistant: only resistant to a single first-line anti-TB drug [20]. Polydrug-resistant: resistant to multiple first-line anti-TB drug excluding both INH and RIF [20]. Multidrug-resistant (MDR): at least resistant to both INH and RIF at the same time [20]. Any resistant: resistant to at least one drug.

### 2.3. Principle of PCR-Reverse Membrane Hybridization

PCR is performed on MTB DNA using specific primers. The amplification products are labeled with Biotin and subjected to molecular hybridization with probes on the membrane strip, then combined with streptavidin coupled peroxidase by Biotin. The color reaction of 3,3,5′,5′-tetramethylbenzidine (TMB) is catalyzed by hydrogen peroxide.

### 2.4. Mycobacterium Species Identification

8–10 FFPE specimen pieces were cut using a 5–10 μm thickness (RM22335, Leica Microsystems Trading Corporation, Wetzlar, Germany). After dewaxed, and lysed, digested with enzymes, the DNA was extracted (QIAamp DNA FFPE Tissue Kit, QIAGEN Corporate Management GmbH, Dusseldorf, Germany). Then DNA was added to the PCR tube (*mycobacterium* species identification gene detection kit, Yaneng Biotechnology Co., Ltd, Shenzhen, China), and amplified with the procedure (s1000, Bio-Rad Life Medical Products Co., Ltd, Hercules, USA): 50 °C for 2 min, 95 °C for 10 min, 95 °C for 45 s, 68 °C for 60 s × 30 times, 95 °C for 30 s, 54 °C for 30 s, 68 °C for 60 s × 30 times, and 68 °C for 10 min. Membrane strips were loaded into a 15 mL tube containing 5–6 mL of liquid A mixed with amplification product. The mixture was heated in a water bath kettle for 10 min and hybridized at 59 °C for 1.5 h (YN-H16, Yaneng Biotechnology Co., Ltd, Shenzhen, China). Place membrane strip in a 50 mL tube preheated to 59 °C containing 40 mL of liquid B, and then wash at 59 °C for 15 min. Membrane strip were incubated in liquid A containing POD enzyme (POD enzyme: liquid A = 1:2000) for 30 min and washed with liquid A and C for 5 min, respectively. The membrane strip was placed in the chromogenic solution (19 mL C +1 mL TMB + 10 μL 3% hydrogen peroxide) and the light was avoided for 10 min. Stopped the reaction with pure water and observed the results. The presence of blue dots represented the detection of the site. Positive and negative controls were established for each experiment. The sequence of detection site on the membrane strip is shown in Figure 1.

### 2.5. Detection of Mutation in MTB Drug Resistance-Related Gene

The process is the same with *Mycobacterium* species identification, except the PCR tube and membrane strip (*Mycobacterium tuberculosis* rifampicin resistance mutation gene detection Kit, Yaneng Biotechnology Co., Ltd, Shenzhen, China). Positive and negative controls were set for each experiment. The sequence of detection site on the membrane strip is shown in Figure 2. More mutation types for membrane strips are in the Appendix A.

### 2.6. Histological Staining

Paraffin sections were stained with H&E to observe basic pathological degeneration. Acid-fast staining was performed to detect positive bacilli. Briefly, bake the slide at 72 degrees for 30 min, xylene for 10 min × 2 times dewaxing, gradient absolute alcohol for 5 min × 3 times. Basic fuchsin stains 2 h. Decolorization with 1% hydrochloric acid alcohol for 5 s. Observe the color until light pink, wash with water. Stain with hematoxylin solution for 30 s. After gradient absolute alcohol rapid dehydration and xylene transparent for 3 times, the slide was sealed with neutral resin. The specific procedure for H&E staining is in the Appendix A.

### 2.7. Statistical Analysis

Photoshop CS6 (Adobe Systems Incorporated, San Jose, USA) used to process pictures. GraphPad Prism 9 (GraphPad Software, San Diego, USA) used to make graphics, and SPSS Statistics 20.0 (International Business Machines Corporation, Amunk, USA) used to collate and analyze the data. The counting data were expressed in cases (*n*) and constituent rate (%). K-S test was used for normality test. The measurement data conforming to normal distribution were statistically described by ($\bar{X}$ ± s), and otherwise were described by P_50_ (IQR). Chi-square test (*χ*^2^) was used, and *p* < 0.05 was statistically significant.

## 3. Results

### 3.1. Histological Features

After infection with MTB, the most typical histopathological change is the formation of a tuberculous granuloma. Tuberculous granulomas are of diagnostic significance. The granuloma has a central caseous necrosis surrounded by epithelioid cells with Langerhans giant cells scattered within it, and lymphocytes accumulate at the periphery of the granuloma. Microscopically, all samples showed the typical pathological features of MTB infection, that is, chronic granulomatous inflammation with a small amount of caseous necrosis (Figure 3A), and acid-fast staining detected positive bacteria (Figure 3B), represented by lung tissue. All these results prove that the specimens are MTB-positive.

### 3.2. Drug Resistance and Molecular Characteristics of MTB

Among 380 cases of MTB, 85 (22.37%) were susceptible to four anti-TB drugs and the remaining 295 (77.63%) were resistant to one or more drugs. The resistant specimens included 93 monodrug-resistant (24.47%), 155 MDR (40.80%), and 47 polydrug-resistant (12.37%). The rate of MDR-TB was higher in previously treated cases (52.53%) than in new cases [(36.65%), *p* < 0.05]. The *p*-value scale * represents values less than 0.05 and is statistically significant (Table 2).

Of the MTB specimens in the study, 55.53% (211/380) were obtained from males. Among specimens obtained from males, 73.46% (155/211) were new cases and 26.54% (56/211) were previously treated cases, and the rate of drug resistance was higher in new cases than in previously treated cases (*p*  <  0.05). Among specimens obtained from females, 25.44% (43/169) were previously treated cases. Of previously treated cases in the study, the rate of drug resistance was higher in females than in males (*p*  <  0.05). The *p*-value scale * represents values less than 0.05 and is statistically significant. (Table 3).

Of all patients with drug-resistant MTB, only two were older than 80 years. The case of specimens resistant to one or more drugs ranged from 2 (> 80 years old) to 120 (18–40 years old). No susceptible specimens and polydrug-resistance specimens were gained from patients older than 80 years. The case of MDR-TB ranged from 1 (> 80 years old) to 60 (18–40 years of age) (Figure 4).

Among 380 cases of MTB, *rpoB* gene exhibited 22 different types of mutated forms. The majority (53/380, 13.95%) of *rpoB* gene carried the D516V mutation. Of the total specimens, 57.63% (219/380) featured the 315M mutation in the *katG* gene, and 13.42% (51/380) featured mutations in both the *katG* and *inhA* genes. Among the total specimens, 18.68% (71/380) carried the 88 M mutation in the *rpsL* gene, and the *embB* gene focused on the 306 M2 mutation with a mutation rate of 19.74% (Table 4).

The mutation rate of the D516V was the highest at 2.45% in monodrug-resistant of RIF, and 86.5% in resistance to more than one drug, respectively. No mutation of H526D, S531L, and S531W occurred in RIF monodrug-resistant. Among the resistant INH, the mutation rate of −15 M was higher in resistance to more than one drug than in monodrug-resistant (*p*  <  0.05). The 43 M and 88 M mutations exhibited the same rate in STR monodrug-resistant. Only two specimens featured 306 M2 mutation in monodrug-resistant of EMB. The *p*-value scale * represents values less than 0.05 and is statistically significant (Table 5).

## 4. Discussion

MTB can infection almost any anatomical site. If MTB is found in the sputum, urine, cerebrospinal fluid, pleural fluids or FFPE specimen, the diagnosis is simple [21]. The final diagnosis of pleural tuberculosis and pulmonary tuberculosis is currently made by demonstrating the presence of tuberculosis bacteria in samples such as sputum/pleural liquids and/biopsy or by examining the tissue for granulomas histologically [22]. However, the insufficient volume and quality of the sputum often hinder the improvement of diagnostic rate, especially when some patients could not provide sputum samples because of low sputum production or difficulty coughing [23]. The majority of tuberculous pleural effusions are exudates with high levels of adenosine deaminase (ADA), lymphocyte-rich, and free flowing, with a low production of MTB culture [21]. FFPE specimen is not as readily available as sputum and pleural fluid, especially for extrapulmonary tuberculosis (EPTB). Spinal tuberculosis is a very hazardous form of skeletal tuberculosis because it can be related to neurological dysfunction due to compression of contiguous neural structures and severe spinal deformity [24]. The Diagnosis of EPTB is often delayed due to the atypical clinical presentation of EPTB. It is often necessary to obtain tissue sample for diagnosis and management of complications through surgery or by the use of computerized tomography (CT), magnetic resonance imaging (MRI), and endoscopy to greatly assist in the anatomic localization of EPTB [25]. FFPE sample can be stored for years, extracting MTB DNA from samples in a short period of time, but require additional deparaffinization steps resulting in partial degradation of the extracted DNA [26]. FFPE specimen is an important pathological specimen for the diagnosis of TB [27]. TB is characterized pathologically by granulomatous inflammation, which is typically consist of central caseous necrosis and surrounding fibrocytes and lymphocytes [28]. Previous studies have confirmed that FFPE specimen has excellent performance in the molecular diagnosis of DR-TB [29,30].

The selection of tuberculosis treatment is often dependent on culture-based phenotypic drug susceptibility tests (pDST). pDST is the gold standard method for the detection of DR-MTB, and it has been widely used for over 50 years [2], but it is time-consuming, technically challenging and requires advanced laboratory facilities [31,32]. Such disadvantages could result in missing out on the opportunity for optimal treatment and transmission of DR-MTB strains, especially in low and medium income countries. The PCR-reverse membrane hybridization technique also has its limitations. Only mutations in resistance-associated genes of first-line drugs are detected, and the number of genes and mutations detected is limited. That is, 85 of the 380 specimens with complete sensitivity may have additional mutations in resistance-related genes that were not detected. DNA sequencing analysis technology can be used for accurately sequence specimens, but it is not affordable for low-income people. By comparison, PCR amplification technology requires lesser amount of target DNA, and then reverse hybridization is performed with specific probes on the membrane strip to directly detect MTB DNA extracted from FFPE specimens. Testing time reduced to 1 day, which provides a new insight for the detection of MTB drug resistance.

In our study, the overall rate of drug resistance was high. This may be due to the fact that the Eighth Medical Center of Chinese PLA General Hospital is a designated TB hospital, which has accepted many patients who have been treated repeatedly, including many patients who have not been diagnosed TB for several months treatment at other hospitals, resulting in a generally high level of drug resistance. Among the total specimens, 4 cases were RIF monodrug-resistant that is rare, which was consistent with previous report [33]. The total rate of RIF resistance (163/380, 42.89%) was higher than the rate in the national epidemiological survey data published in 2010 (7.5–33.5%) [34]. However, it could be influenced by different regions and genetic strains. In a study that occurred in Guangzhou Chest Hospital of China, 85.71% were RIF-resistant, of which 38% were MDR-TB [35]. A study from Myanmar showed 62% isolates had RIF resistance [36]. Between 85–90% of MTB has been reported that resistance to INH when it has the RIF resistance, and RIF resistance is regarded as a major marker of MDR [37]. The most mutations in *rpoB* gene concentrated on codon 516, which was inconsistent with relevant studies that codon 531 was more frequently [38,39,40]. However, a study in Brazil showed that codon 516 *rpoB* mutations exhibited a higher frequency than codon 531 mutations in Haarlen lineage [41]. It has been reported that the MTB drug resistance-associated genes are polymorphic, that endemic bacterial flora differ in different regions, and that mutation hot spots do not coincide [42]. The mutation rate of 315 M in the *katG* gene was 78.49% (219/279), which was higher than 56.22% reported in relevant report [43]. Further, 3.22% (9/279) carried the −15 M mutation in the *inhA* gene, which is close to 3.7% in relevant study [44] and lower than 10.10% in relevant report [43]. In this study, the rate of 43 M mutation is lower than 88 M. It is somewhat inconsistent with the higher mutation rate of 43 M reported [45]. However, mutation rate of 43 M accounted for 45.80% of the total STR resistance, which was close to 53.5% reported in the literature [46]. Moreover, 47.74% (74/155) of MDR-TB is EMB-resistant, lower than 51.3% to 66.7% of MDR-TB was EMB-resistant in some parts of China [47]. It has been reported that the condon 306 *embB* mutations are inconsistent with EMB resistance. The mutations of codon 306 in the *embB* gene could not directly cause the EMB resistance, but it easily leads to resistance to any drugs [19].

The genetic background of the host influences the manifestation of TB, and MTB can infect a large number of susceptible people [48]. The progression from MTB infection to disease and from disease to death depends on a variety of factors, with age being central to all of these transitions [49]. There is growing recognition that TB is an important preventable cause of childhood morbidity and mortality in areas where tuberculosis is endemic [50]. It is estimated one million children and adolescents develop TB each year, and about 226,000 of whom die [51]. Meanwhile, 30,000 children become ill with MDR-TB each year [52]. Similar to EPTB, child TB is not straightforward to detect and diagnose [53]. Strengthening clinical and laboratory diagnostics, including MDR-TB, and providing recommended protocols for disease and infection treatment is an urgent necessity [54]. In a study of risk factors for secondary TB, younger age, human immunodeficiency virus (HIV), and low weight were found to be risk factors for children, while for women, HIV, and low weight were risk factors, and for men, HIV and low body mass index led to more rapid development of TB [55]. The prevalence of TB is significantly higher in men than in women, especially in low- and middle-income countries [56]. Nevertheless, the gender differences varied widely geographically [57].

The most of DR-MTB are primary resistance instead of acquired resistance [58]. The latter refers to the resistance obtained during the course of treatment. Person-to-person transmission of DR-MTB strains is the main mechanism of drug resistance in MTB [58,59]. It has also been reported that the current burden of drug-resistant TB is driven by the evolution of several mechanisms together, including sustained transmission and intra-patient drug resistance [60]. A study from Beijing of China showed that the detection rate of MDR-TB was higher in previously treated patients than in newly diagnosed TB patients (34.5% versus 6.8%) [10]. In a study from Shandong Province in China, a decreasing trend in overall drug resistance among new TB cases was found from 2004 to 2018, but primary drug resistance patterns are shifting from female, INH-resistant TB to male, RIF/STR-resistant TB [58]. Meanwhile, it suggests that in the future more attention needs to be given to women, smoking, alcohol consumption, or TB subgroups aged 15 to 44 years, in order to control primary DR-TB.

Over the past few years, the worldwide burden of MDR-TB has risen by more than 20 percent annually, resulting in an estimated that DR-TB will lead to 75 million deaths in the next 35 years [61]. The spread of DR-TB is so rapid especially because of inferior TB drug regimens and the presence of DR-TB in high-risk patients such as HIV patients or in high-risk settings such as hospitals [62]. It is necessary to quickly diagnose and administer rapid anti-TB treatment for TB in order to reverse the morbidity associated with TB [22]. MDR-TB will soon develop drug resistance when treated with other first-line drugs, and second-line anti-TB drugs will take longer to treat and will be more cost-effective, toxicity-intensive, and expensive than first-line drugs [63]. The increase in resistance to MTB is a serious problem both for developing countries and for developed countries [64]. Although RIF, INH, STR, and EMB are the first-line drugs in clinical treatment for MTB, the single drug does not achieve satisfactory therapeutic effects. Drug combination therapy can promote the absorption of injuries and effectively treat TB [65]. However, the development of DR-MTB reduces clinical therapeutic effects, increases clinical treatment difficulty, which is not conducive to the recovery of patients and increases the risk of transmission and epidemic [66]. Therefore, detection of drug resistance of MTB timely is helpful for the adjustment of therapeutic schedule, the rational use of medicines, and the improvement of therapeutic effects.

## 5. Conclusions

Drug resistance is a key factor affecting the treatment of TB, and its characteristics change over time. The current TB burden remains severe, especially the increase in DR-TB, which represents a growing challenge to public health. DR-TB delays diagnosis and prolongs treatment, resulting in adverse outcomes for patients. Understanding the drug resistance and molecular characteristics of MTB is beneficial for clinical treatment and control of TB transmission.

## Figures and Tables

**Figure 1 jpm-12-02088-f001:**
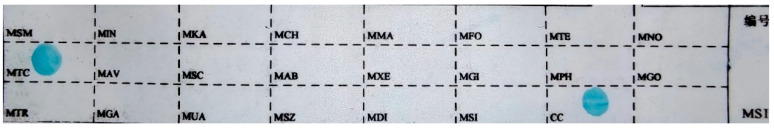
Sequence of detection site on membrane strip. MTC, *Mycobacterium tuberculosis* complex; CC, quality control site; the rest were *non-tuberculous mycobacterium* detection sites. The Chinese in the upper right corner means that you can write numbers in this area to number the membrane strips.

**Figure 2 jpm-12-02088-f002:**
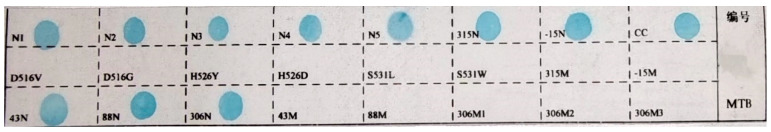
Sequence of detection site on membrane strip. *rpoB* gene [D516V, aspartate (D)→valine (V), a A→T mutation at nucleotide position 1547; D516G, aspartate (D)→glycine (G), a A→G mutation at nucleotide position 1547; H526Y, histidine (H)→tyrosine (Y), a C→T mutation at nucleotide position 1576; H526D, histidine (H)→aspartate (D), a C→G mutation at nucleotide position 1576; S531L, serine (S)→leucine (L), a C→T mutation at nucleotide position 1592; S531W, serine (S)→tryptophan (W), a C→G mutation at nucleotide position 1592]; *katG* gene (315 M, a G→C or G→A mutation at nucleotide position 944); *inhA* gene (−15 M, a C→T mutation at nucleotide position −15); *rpsL* gene (43 M, a A→G mutation at nucleotide position 128; 88 M, a A→G mutation at nucleotide position 263); *embB* gene (306 M1, a G→H mutation at nucleotide position 918; 306 M2, a A→G mutation at nucleotide position 916; 306 M3, a A→C mutation at nucleotide position 916). The Chinese in the upper right corner means that you can write numbers in this area to number the membrane strips.

**Figure 3 jpm-12-02088-f003:**
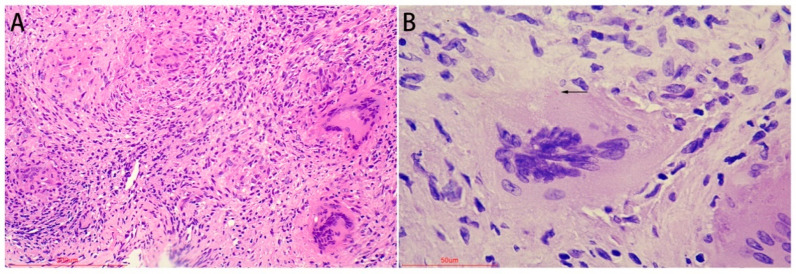
Histological features represented by lung tissue. H&E staining (**A**), tuberculous granulomas with caseous necrosis; Acid-fast staining (**B**), the black arrow showed the positive bacteria.

**Figure 4 jpm-12-02088-f004:**
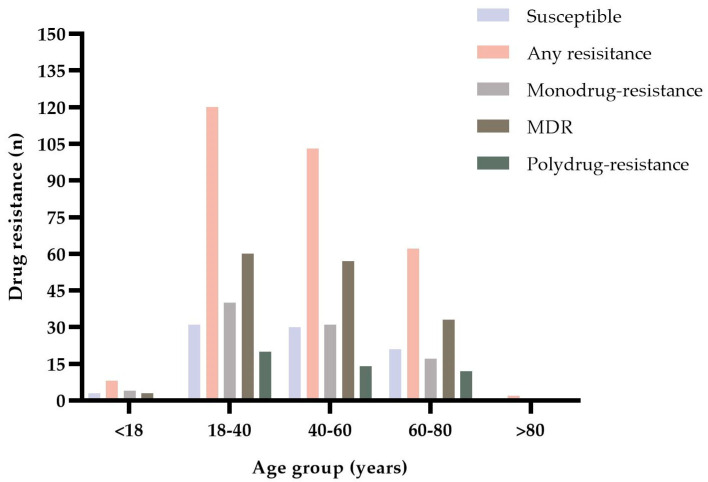
Number of 380 MTB-positive FFPE specimens by age group and drug resistance pattern.

**Table 1 jpm-12-02088-t001:** Specimen basic information.

Parameter		MTB-Positive FFPE Specimens
Age		43.00 (29)
	≥60/<60	68.00 (10)/36.00 (21)
Gender	male/female	211/169
Tissue type	lung/kidney/liver/intestine/brain/thyroid	39/57/2/8/3/1
	spine/arthrosis	172/45
	lymph node	13
	skin	2
	pericardium/pleura/peritoneum	2/25/11
Treatment type	new cases/previously treated cases	281/99
Region	the north of China/the southeast of China	278/36
	the Central and western of China	66

**Table 2 jpm-12-02088-t002:** Drug resistance of 380 MTB-positive FFPE specimens.

Drug Resistance	New Cases	Previously Treated	Total	*χ* ^2^	*p*
	*n* = 281	Cases (*n* = 99)	*n* = 380		
	*n*, %	*n*, %	*n*, %		
Susceptible	68, 24.20	17, 17.17	85, 22.37	2.082	0.149
Any resistant	213, 75.80	82, 82.83	295, 77.63	2.082	0.149
Monodrug-resistant	74, 26.33	19, 19.19	93, 24.47	2.021	0.155
resistance to RIF	3, 1.07	1, 1.01	4, 1.05		
resistance to INH	65, 23.13	18, 18.18	83, 21.84		
resistance to STR	4, 1.42	0, 0	4, 1.05		
resistance to EMB	2, 0.71	0, 0	2, 0.53		
MDR	103, 36.65	52, 52.53	155, 40.80	7.635	0.006 *
resistance to RIF + INH	32, 11.39	12, 12.12	4, 11.58		
resistance to RIF + INH + STR	22, 7.83	15, 15.15	37, 9.74		
resistance to RIF + INH + EMB	15, 5.34	3, 3.03	18, 4.74		
resistance to RIF + INH + STR +EMB	34, 12.10	22, 22.22	56, 14.74		
Polydrug-resistant	36, 12.81	11, 11.11	47, 12.37	0.021	0.884
resistance to RIF + STR	2, 0.71	0, 0	2, 0.53		
resistance to RIF + EMB	1, 0.36	0, 0	1, 0.26		
resistance to RIF + STR +EMB	1, 0.36	0, 0	1, 0.26		
resistance to INH + STR	12, 4.27	4, 4.04	16, 4.21		
resistance to INH + EMB	7, 2.49	5, 5.05	12, 3.16		
resistance to INH + STR + EMB	11, 3.91	2, 2.02	13, 3.42		
resistance to STR + EMB	2, 0.71	0, 0	2, 0.53		

**Table 3 jpm-12-02088-t003:** Cross analysis table of treatment type, gender and drug resistance.

Parameter		Males	Females	*χ* ^2^	*p*
New cases	Susceptible	36	32	0.179	0.673
	Any resistance	119	94		
Previously treated cases	Susceptible	5	12	6.610	0.013 *
	Any resistance	51	31		
*χ* ^2^	-	5.371	0.105	-	-
*p*	-	0.020 *	0.746	-	-

**Table 4 jpm-12-02088-t004:** Mutations in drug resistance-related genes.

Drug and Gene	Mutation Form	Total (*n* = 380)	
		*n*	%
RIF, *rpoB*	D516V	53	13.95
	D516G	2	0.53
	H526Y	1	0.26
	S531L	6	1.58
	D516V + D516G	24	6.32
	D516V + H526Y	3	0.79
	D516V + H526D	11	2.89
	D516V + S531L	1	0.26
	D516V + D516G + H526D	17	4.47
	D516V + D516G + S531L	1	0.26
	D516V + D516G + H526Y + H526D	9	2.37
	D516V + D516G + H526Y + H526D + S531L	8	2.11
	D516V + D516G + H526Y + H526D + S531L + S531W	7	1.84
	D516V + D516G + H526D + S531L	5	1.32
	D516V + H526Y + H526D	2	0.53
	D516V + H526Y + H526D + S531L	1	0.26
	D516V + H526D + S531L	3	0.79
	D516G + H526Y	1	0.26
	D516G + H526D	2	0.53
	H526Y + H526D	3	0.79
	H526Y + H526D + S531L	1	0.26
	H526D + S531L	2	0.53
INH, *katG*	315 M	219	57.63
*inhA*	−15 M	9	2.37
	−15 M + 315 M	51	13.42
	43 M	28	7.37
STR, *rpsL*	88 M	71	18.68
	43 M + 88 M	32	8.42
	306 M1	6	1.58
	306 M2	75	19.74
EMB, *embB*	306 M1 + 306 M2	18	4.74
	306 M2 + 306 M3	1	0.26
	306 M1 + 306 M2 + 306 M3	5	1.32

D, aspartic; V, valine; G, glycine; H, histidine; Y, tyrosine; S, Serine; L, leucine; W, tryptophan; D516V, A→T; D516G, A→G; H526Y, C→T; H526D, C→G; S531L, C→T; S531W, C→G; 315 M, G→C or G→A; −15 M, C→T; 43 M, A→G; 88 M, A→G; 306 M1, G→H; 306 M2, A→G; 306 M3, A→C.

**Table 5 jpm-12-02088-t005:** Mutation sites in monodrug-resistant and resistance to more than one drug.

Drug and Total Case of Resistace	Mutation Sites	Mutation in Monodrug-Resistant	Mutation in Resistance to More than One Drug	*χ* ^2^	*p*
*n*		*n*, %	*n*, %		
RIF, 163	D516V	4, 2.45	141, 86.5	0.509	0.476
	D516G	1, 0.61	75, 46.01	0.771	0.380
	H526Y	1, 0.61	35, 21.47	0.020	0.887
	H526D	0, 0	71, 43.56	3.165	0.075
	S531L	0, 0	35, 21.47	1.121	0.290
	S531W	0, 0	7, 4.29	0.184	0.668
INH, 279	315 M	81, 29.03	189, 67.74	0.252	0.616
	−15 M	5, 1.79	55, 19.71	16.775	0.000 *
STR, 131	43 M	2, 1.53	58, 44.27	0.029	0.864
	88 M	2, 1.53	101, 77.10	2.012	0.156
	306 M1	0, 0	29, 27.62	0.778	0.378
EMB, 105	306 M2	2, 1.90	97, 92.38	0.124	0.725
	306 M3	0, 0	6, 5.71	0.124	0.725

D, aspartic acid; V, valine; G, glycine; H, histidine; Y, tyrosine; S, Serine; L, leucine; W, tryptophan; D516V, A→T; D516G, A→G; H526Y, C→T; H526D, C→G; S531L, C→T; S531W, C→G; 315 M, G→C or G→A; −15 M, C→T; 43 M, A→G; 88 M, A→G; 306 M1, G→H; 306 M2, A→G; 306 M3, A→C.

## Data Availability

Data available from the author upon request.

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
