# Peer review of "Drug Resistance and Molecular Characteristics of Mycobacterium tuberculosis: A Single Center Experience"

_jpm, 2022, doi:10.3390/jpm12122088_

Round 1
Reviewer 1 Report
Manuscript deals with the analysis of drug resistance and molecular characteristics of Mycobacterium Tuberculosis (MTB), using a total of 380 MTB positive cases diagnosed in a single hospital.
They identified mutations on the following genes, the rifampicin (RIF) resistance gene rpoB, the isoniazid (INH) resistance gene inhA, the streptomycin (STR) rpsL resistance gene, and the embB gene of ehambutol (EMB) resistance.
They have used PCR amplification technology on a small amount of target DNA, and then reverse hybridization with specific probes on the membrane strip to directly detect MTB DNA extracted from FFPE specimens.
This technology seems to be specific and detection time has been shortened to 1 day, which provides a new idea for the rapid detection of MTB drug resistance. Results are sounds.
Nevertheless, more information about the specific probes on the membrane strip must be provided in the material and methods section or as supplmentary materials.
Author Response
Dear reviewer:
Thank you very much for reviewing our manuscript and giving us the opportunity to revise it. We take it very seriously and have already made changes to the manuscript based on your suggestions. The attachment below is a point-by-point response to your comments. Please see the attachment.
Sincerely Yours,
Prof. Fenghua Wang
Department of Pathology, the Eighth Medical Center, Chinese PLA General Hospital, Beijing 100091, China
E-mail: 13810915901@163.com

Reviewer 2 Report
Scientific names need to be italicized and follow the rules. Review the text, making corrections starting with the title.
In the abstract (line 8), the authors say “...incidence and mortality of Mycobacterium tuberculosis”. Wouldn't it be incidence of tuberculosis and mortality caused by the disease? Fix it, please.
The authors always mention “our hospital” to indicate the origin of the samples. It is necessary to correctly indicate the hospital and the institution/laboratory/department.
The kit mentioned in line 52 needs to be better specified. In addition, it is also necessary to explain, more clearly, how the kit identifies mycobacteria species and, mainly, resistance genes. Still in this context, from the description of the kit, it is only possible to identify genes resistant to rifampicin, however the results show resistance genes to other drugs.
The sub-topic “Drug resistance pattern” in the “Materials and methods” section needs to be better explained. What did the authors mean by this subtopic? Wouldn't it be better to include it elsewhere in this section?
In the sub topic “Statistical analysis” the N sample and percentage are explained. This is not statistical analysis. It is necessary to make real statistical analyzes of the results and explain how they were done.
The authors did not mention the ethical aspects of the work, since patient samples were used. Even if it is waived, this part needs to be explained.
The Histological features shown in the results need to be better explained. Indicate what these images are really showing and what is the importance of this data for the work.
The authors need to better defend the results and be clearer on the central question of the work. The interpretation of the results in the Discussion section was not well done, since the authors only repeated the description of the obtained data for the most part. Furthermore, the authors need to convince the scientific community how this study brings some development to the area.
Author Response

(The authors gave the same response as above.)

Reviewer 3 Report
Reviewer’s Comment
The article by Li et al on “Drug Resistance and Molecular Characteristics of Mycobacterium Tuberculosis: A Single Center Experience” discuss about the drug resistance and molecular characteristics of MTB to the four first-line drugs. The polymerase chain reaction (PCR)-reverse membrane hybridization technology to test 380 MTB-positive formalin-fixed and parrffin-embedded tissue (FFPE) specimens was used and results were summarized and analyzed for drug resistance and molecular characteristics. The paper needs major modification, before considering it for acceptance.
Comments
1. It is not clear from the data that among the samples included how many were on re-treatment. It is very important, as observed high prevalence can be due to inclusion of re-treated cases in large number.
2. It is also not clear, how the patients were confirmed for TB positivity. Also as per treatment guideline, the DST is performed, before the initiation of treatment. If the data is available concordance with observed results should be presented.
3. The demographic characteristics of the included patients should be presented in the paper.
4. The result does not discuss about the drug resistance profile in different genders and age group. These associations are important because of differential susceptibility to TB.
5. What samples have been included in the study is not mentioned. Author has only mentioned that 380 MTB-positive formalin-fixed and parrffin-embedded tissue (FFPE) specimens, but tissue type is not specified.
6. English is very poor and need extensive editing.
Minor comments
Line 113: Author has written “We mainly detected at codon…….”. But what detected is not cleared.
Line 124-25: Sentence need to be rewritten.
Line 135-137: Very poor English. What is its advantage need to be written properly?
Line 160-161: Author have written “The majority of RIF resistance gene mutations were at codon 516, which was inconsistent with relevant studies that more frequently mutations were at condon 531” Author cited only one reference while he is using the relevant studies terminology.
7. In Discussion section, author has not discussed about related studies done within the Country in nearby area and how their results varies/similar with present studies.
8. Conclusion: It is very short and need to be elaborated and improved.
Author Response

(The authors gave the same response as above.)

Round 2
Reviewer 2 Report
The authors made all the necessary modifications, showing a significant improvement in writing, which facilitated the understanding of the manuscript and its relevance to the scientific area.